# Sex Determination and Differentiation in Decapod and Cladoceran Crustaceans: An Overview of Endocrine Regulation

**DOI:** 10.3390/genes12020305

**Published:** 2021-02-21

**Authors:** Kenji Toyota, Hitoshi Miyakawa, Chizue Hiruta, Tomomi Sato, Hidekazu Katayama, Tsuyoshi Ohira, Taisen Iguchi

**Affiliations:** 1Marine Biological Station, Sado Center for Ecological Sustainability, Niigata University, Sado, Niigata 952-2135, Japan; 2Department of Biological Sciences, Faculty of Science, Kanagawa University, Hiratsuka, Kanagawa 259-1293, Japan; ohirat-bio@kanagawa-u.ac.jp; 3Department of Biological Science and Technology, Faculty of Industrial Science and Technology, Tokyo University of Science, Katsushika, Tokyo 125-8585, Japan; 4Center for Bioscience Research and Education, Utsunomiya University, Utsunomiya, Tochigi 321-8505, Japan; h-miya@cc.utsunomiya-u.ac.jp; 5Department of Biological Sciences, Faculty of Science, Hokkaido University, Sapporo, Hokkaido 060-0810, Japan; chizueh@gmail.com; 6Graduate School of Nanobioscience, Yokohama City University, Yokohama, Kanagawa 236-0027, Japan; 7Department of Applied Biochemistry, School of Engineering, Tokai University, Kanagawa 259-1292, Japan; katay@tokai-u.jp

**Keywords:** crustacean, sex determination, sexual differentiation, insulin-like androgenic hormone (IAG), androgenic gland hormone (AGH), crustacean female sex hormone (CFSH), juvenile hormone (JH)

## Abstract

Mechanisms underlying sex determination and differentiation in animals are known to encompass a diverse array of molecular clues. Recent innovations in high-throughput sequencing and mass spectrometry technologies have been widely applied in non-model organisms without reference genomes. Crustaceans are no exception. They are particularly diverse among the Arthropoda and contain a wide variety of commercially important fishery species such as shrimps, lobsters and crabs (Order Decapoda), and keystone species of aquatic ecosystems such as water fleas (Order Branchiopoda). In terms of decapod sex determination and differentiation, previous approaches have attempted to elucidate their molecular components, to establish mono-sex breeding technology. Here, we overview reports describing the physiological functions of sex hormones regulating masculinization and feminization, and gene discovery by transcriptomics in decapod species. Moreover, this review summarizes the recent progresses of studies on the juvenile hormone-driven sex determination system of the branchiopod genus *Daphnia*, and then compares sex determination and endocrine systems between decapods and branchiopods. This review provides not only substantial insights for aquaculture research, but also the opportunity to re-organize the current and future trends of this field.

## 1. Introduction

Crustaceans form a large subgroup of arthropods that live in virtually all regions of Earth. The latest molecular phylogenetic studies of arthropods have revealed that extant Crustacean lineages can be categorized into three major groups: Ostracoda, such as the sea firefly, Malacostraca such as crabs and shrimps, and Branchiopoda such as water fleas and brine shrimp; and that the Crustacea and Hexapoda (insect group) together form the Pancrustacea [1,2,3] (Figure 1). The sex manipulation of Malacostraca, especially species that are important for fishery, has been more thoroughly studied than those of other crustaceans, since it has been recognized as an effective and beneficial technique for aquaculture. Generally, fishery-important Malacostracans have different commercial values between females and males, due to differences in growth rates between sexes and animals of larger body size being of higher value. Additionally, in terms of building up large-scale aquaculture, females may be more valuable than males since they provide more benefit in increasing the numbers of individuals within population [4,5]. In Malacostracans, although the primary sexual fate is generally decided by genetic factors such as the sex chromosomes (genotypic sex determination: GSD), androgenic gland factor and crustacean female sex hormone (CFSH) are also recognized as the peptide hormones involved in the development of sexually dimorphic characteristics [6,7]. For Branchiopoda, it is known that the majority has GSD but not peptides such as insulin-like androgenic gland hormone (IAG) [8] and CFSH in their genomes. Furthermore, part of the Branchiopoda, such as the cladoceran water flea genus *Daphnia*, have environmental sex determination (ESD). Recently, understanding of the molecular mechanisms underlying sex determination and differentiation in daphnids has been enhanced by the discovery of the phenomenon that induces male-biased production in response to juvenile hormone (JH) exposure. In this review, we summarize current knowledge on sex determination mechanisms and sex hormones (IAG and CFSH) in Malacostraca decapods and JH-driven sex determination pathways in the Branchiopod cladocerans from various studies, including recently developed OMICS approaches.

## 2. Sex Determination and Differentiation Mechanisms in Crustaceans

Sex determination is the most fundamental developmental process that governs the establishment of sexually dimorphic traits, and then leads to sex-specific characteristics in physiology and behavior. Although development as either a female or male is a robust mechanism in animals, there is an amazing diversity of modes of sex determination. In most organisms, sexual fate is thought to be genetically pre-decided at fertilization (GSD) rather than to be determined by environmental cues (ESD). Substantial examples of GSD factors are sex chromosomes that carry sex-determining genes. Such sex chromosome systems can be grouped into two major forms: male heterogamety (called XX/XY system), and female heterogamety (called ZW/ZZ system). In Malacostraca, the majority of shrimps, crayfishes, and terrestrial isopods employ a ZZ/ZW sex determination system [9,10,11,12,13,14,15], while some species of crabs and lobsters employ XX/XY determination [16,17,18,19] (Figure 2). Mode of sex determination in decapods, isopods, amphipods, and branchiopods is summarized in Table 1 and has been well reviewed [20].

Sexual differences arise during embryogenesis, even though the genomic content differs little between females and males. Thus, differences in gene regulation are generally considered to underlie most of the sex-specific differentiation, and many researchers have therefore, sought to identify regulatory mechanisms that govern sex-specific gene expressions.

The molecular cascades leading to distinct sexual phenotypes are triggered by a wide variety of genetic or environmental factors, however, most of them tend to converge on a common set of transcriptional regulators. Such transcriptional factors are doublesex (dsx) and male-abnormal-3 (DM) domain-containing genes [41,42]. The first identified DM domain-containing gene was the dsx from the fruit fly *Drosophila melanogaster*, named for its importance in both female and male development [43]. In insect species, *dsx* genes play a pivotal role in sexual differentiation, and are involved in the formation of sexually dimorphic traits through the expression of sex-specific isoforms [41,42]. Likewise, the DM domain-containing genes have been implicated in the determination and/or maintenance of gonadal sex across a broad range of vertebrate species, such as the Y-chromosome-linked *DMY* gene in the medaka fish [44], the W-chromosome-associated *DM-W* gene in the African clawed frog *Xenopus laevis* [45], and Z-chromosome-linked *DMRT1* gene in the chicken *Gallus gallus domesticus* [46]. Unlike in insect species, the dsx function is known to be regulated via sex-biased expression (with the majority of cases in males) rather than alternative sex-specific splicing in non-insect arthropods such as the crustacea (details in following sections) and chelicerata (including the common house spider) [47]. Moreover, in addition to *dsx* genes, the invertebrate Y-chromosome-linked *iDMY* genes have recently been identified as a masculinization factor during embryogenesis in the Eastern spiny lobster *Sagmariasus verreauxi* [48] and in the ornate spiny lobster *Panulirus ornatus* [49]. High-throughput next generation sequencing techniques have successfully enabled the decoding of draft genomes in many Malacostracans (Table 1). Especially genome information of following species: the marbled crayfish *Procambarus fallax f. virginalis* [26], the Pacific white shrimp *Litopenaeus vannamei* [22], the giant freshwater prawn *Macrobrachium rosenbergii* [24], and the terrestrial isopod *Armadillidium vulgare* [29], will accelerate sex determination and differentiation studies, since they have been used in the studies of endocrinology and sex differentiation as experimental animals (details in following sections). Moreover, several transcriptome studies have revealed the existence of female- or male-biased genes in various decapods, shedding light on the understanding of molecular mechanisms underlying sex determination, sexual differentiation, and sexual maturation [48,49,50,51,52,53]. Improving the sequencing depth and algorithm for *de novo* assembly will help to identify the loci of sex-determining genes on sex chromosomes. In terms of ESD, a broad range of abiotic and biotic environmental factors (for example, photoperiod, temperature, social interaction, and parasites) can trigger, both female or male sexual differentiation from a single genotype. A striking example of the ESD system in crustaceans is the Branchiopoda cladoceran water flea *Daphnia* (reviewed in following Section 5) (Figure 2).

As a topic of growing concern over environmental contamination by human activity, the impacts of chemical pollution on living organisms are no longer negligible. Although we will describe this in more detail in the following Section 6, it is already known that sex determination and/or sexual differentiation processes in various crustacean orders can be disrupted by endocrine disrupting chemicals (EDCs) such as in human sewage (e.g., detergents and medicines), pesticide residues, and heavy metals [54].

## 3. Androgenic Gland Factors

The integrated signaling cascades responsible for sexual differentiation are almost as diverse, ranging from cell-nonautonomous gonad-dependent endocrine control (mainly by sex steroids such as estrogens and androgens) of sexual traits in mammals and other vertebrates to cell-autonomous sex determination in invertebrates such as insects [41]. However, exceptionally, only Malacostracan crustaceans have a cell-nonautonomous sexual differentiation manner and, unlike gonad-dependent endocrine regulation in vertebrates, have a male-specific endocrine gland known as the androgenic gland (AG), which is located on the terminal section of the vas deferens [55]. The AG has not been described in cladocerans [56]. Briefly, the physiological function of the AG has historically been demonstrated to play a pivotal role in male sex differentiation by AG ablation and implantation in the Malacostracan amphipod *Orchestia gammarella* [55,57]. Later, AG studies have been conducted using the Malacostracan isopod woodlouse *A. vulgare* by AG implantation [58], AG ablation [59], and injections of AG extracts [60]. Thereafter, the androgenic gland hormone (AGH) has been purified, and its peptide structure reported [61,62]. As with the amphipods and isopods, physiological roles of AGH have further been demonstrated in Malacostracan decapod species using, for instances, AG implantation in the red claw crayfish *Cherax quadricarinatus*, and the marbled crayfish *P. fallax f. virginalis* [63], while AG removal from males resulted in feminization in *C. quadricarinatus* [64] and in the freshwater prawn, *M. rosenbergii* [23]. Substantial intrinsic AGH in decapod species has been identified from *C. quadricarinatus*. Further study has demonstrated that the AG hormone structure is very similar to the insulin-like family, and this hormone was termed the IAG [8]. In terms of regulatory mechanisms of IAG expression, some studies have found that eyestalk ablation in males caused hypertrophy and hyperplasia of the AG [65,66] as well as over-expression of the *IAG* gene [67]. Based on those findings, it has been suggested that there is a unique developmental axis known as the X-organ–sinus-gland neuroendocrine complex (XO-SG)-AG-testis axis has been suggested where, some XO-SG-derived neuropeptides act as upstream regulators of *IAG hormone* gene expression [68]. Although the fine details are in dispute, it has been demonstrated that IAG interacts with its binding protein and receptor to activate downstream pathways [69,70,71]. Additionally, recent studies have demonstrated that the *dsx* gene is involved in the regulation of IAG expression. In the Chinese shrimp *Fenneropenaeus chinensis*, the *Fcdsx* gene dominantly expresses in the testis, and the mRNA level is gradually increased with larval development. Knockdown of the *Fcdsx* gene resulted in suppression of *IAG* gene expression, suggesting that Fcdsx regulates male sexual differentiation via IAG signaling [72]. On the other hand, in the red claw crayfish *C. quadricarinatus*, the *Cqdsx* gene mainly expresses in the gonad (two times higher in the ovary than in the testis), and its knockdown increased IAG expression, meaning that Cqdsx is involved in female sexual differentiation [73]. Both Fcdsx and Cqdsx have no sex-specific splicing form and, therefore, there is male- or female-biased expression to promote sexual differentiation pathways. 

A wide range of aspects of IAG has been previously comprehensively overviewed [20,74]. Here, we focus on the relation between the structure and biological activity of IAGs. Although a lot of studies have demonstrated that the silencing of *IAG* genes by RNA interference promotes morphological feminization [75,76,77,78], there is no direct evidence for the function of IAG. The deduced amino acid sequences of IAGs share highly conserved structural features including a signal peptide, B chain, C peptides, and A chain with the mature active peptide formed after removal of the C peptides [79]. As an active form, both A and B chains form a heterodimer with disulfide bonds. Total organic chemical synthesis of IAG has revealed potentially two types of IAG: one is similar to the vertebrate insulin-type, and the other is not an insulin-type (Figure 3). In the isopod *A. vulgare*, our group has found that there are four disulfide bonds and their arrangement is different from that in the vertebrate insulin-type (named as androgenic gland hormone: AGH-type) but it is thermodynamically unstable [80]. Moreover, *in vivo* biological assays demonstrated the AGH-type has the ability to promote masculinization, but the insulin-type does not (Figure 3). As compared with isopods, decapod IAGs lack the two cystein residues found in the isopod AGH, indicating that the decapod IAGs are more related molecularly to the vertebrate insulin [81], although there some exceptions (eight cystein residues as well as isopod species) such as in the Indian bait prawn *Palaemon pacificus* [82,83] and in the freshwater prawn *M. rosenbergii* [77,84]. Moreover, we synthesized both AGH-type and insulin-type IAGs of the kuruma prawn *Marsupenaeus japonicus* by total chemical synthesis and demonstrated that the insulin-type showed a significant biological activity in vitro, whereas the AGH-type did not [81] (Figure 3). This has strongly suggested that the insulin-type IAG is the innate form in the decapod species. In the near future, it will be necessary to prove the *in vivo* functional differences between the insulin-type and AGH-type.

## 4. Crustacean Female Sex Hormone (CFSH)

The CFSH was found to be a responsible factor for regulating the development of female reproductive characteristics in the blue crab *Callinectes sapidus* and the green crab *Carcinus maenas* [7]. Callinectes CFSH, synthesized in the X-organ and then stored in/secreted from the sinus gland, was purified from eyestalk tissues. This discovery of CFSH has resulted in a major research trend for exploring its homologs from other decapod species. To date, eyestalk transcriptome and peptidome approaches have successfully identified CFSH orthologs in several other brachyuran crabs, such as the swimming crab *Portunus trituberculatus* [85], the Chinese mitten crab *Eriocheir sinensis* [86], the green shore crab *C. maenas* [87], and the mud crab *Scylla paramamosain* [88,89,90], as well as in the kuruma prawn *M. japonicus* [91], the Pacific white shrimp *L. vannamei* [86], the banana shrimp *Fenneropenaeus merguiensis* [92], the Antarctic shrimp *Chorismus antarcticus* [93], the Eastern rock lobster *S. verreauxi* [94], the giant freshwater prawn *M. rosenbergii* [86,95,96], the red swamp crayfish *P. clarkii* [85], and the Australian crayfish *C. quadricarinatus* [97]. Despite the growing amount of CFSH sequence information, little is known about its physiological functions. Knockdown of CFSH impaired the development of reproductive traits such as the ovigerous setae, gonopores and extended parental brood care in *C. sapidus* [7], and the formation of gonopores in juvenile stages in the mud crab [89], indicating that CFSH acts as an endocrine factor for establishing female-specific morphological characteristics. However, a few reports have demonstrated that CFSH expression can be detected in both females and males in, for example, the kuruma prawn [98]. Moreover, two distinct CFSH subtypes have been identified from eyestalk and ovary tissues [98]. Based on immunohistochemistry and in situ hybridization analyses of CFSH, the ovary-type is predominantly expressed in oogonia and previtellogenic oocytes during vitellogenesis, indicating that it may take part in reproductive processes. Besides, in the Australian crayfish, CFSH expression has been detected in the central nervous system, antennal gland, and gut [97].

Some recent studies have demonstrated the crosstalk of CFSH with IAG to facilitate sexual differentiation processes. In fact, CFSH has been detected in the eyestalk of both sexes of several crab species [7,88,89]. In the mud crab *S. paramamosain*, a previous study demonstrated that CFSH promotes the formation of female-specific reproductive traits such as gonopores in females, and inhibits the expression of IAG in AG in vitro [88]. Moreover, the machinery of transcriptional regulation of CFSH on IAG expression has been investigated with regard to the involvement of signal transducers and activators of the transcription (STAT)-binding site [89]. Notably, the CFSH receptor has not been identified so far in decapod species. Further studies on the CFSH receptor and its downstream signaling pathways are necessary to understand the mechanisms underlying endocrine crosstalk between CFSH and IAG, and its involvement in sex determination/differentiation in Malacostracans.

## 5. Juvenile Hormone as a Male Sex-Determinant in Cladocerans

Juvenile hormone (JH) is well known as one of the important endocrine factors regulating molting and metamorphosis in insect species. It also shows pleiotropic functions to control various phenomena such as ovarian development, reproductive behavior [99], and various types of phenotypic plasticity such as caste determination in the social insects [100], weapon traits development in the stag beetles [101], and the switching of reproductive modes in the pea aphid [102]. It is currently accepted that the JH system is conserved among Arthropod species [103,104]. In 1987, methyl farnesoate (MF), which is structurally related to insect JHs, was identified as an endogenous JH molecule in the spider crab, *Libinia emarginata* [105]. So far, it has generally been accepted that MF is a major JH in Malacostracan crustaceans [106,107,108]. To date, physiological functions of MF have been demonstrated as stimulation of protein synthesis, promotion of molting cycle, reproduction, and larval development in Malacostracan crustaceans (e.g., crabs and shrimps) given their importance in aquaculture [108,109,110], however, no report is available showing involvement of MF in sex determination and/or sexual differentiation in Malacostracans. 

Within Crustacea, cladocerans belong to the class of Branchiopoda (Figure 1). Cladoceran species, commonly called water fleas, are one of the dominant organisms in freshwater zooplankton communities [111]. The genus *Daphnia* in general employs cyclical parthenogenesis, in which parthenogenesis and sexual reproduction can be altered in response to environmental cues such as day-length, water temperature, nutrition, overcrowding, and their combinations [36,37,112,113]. Under favorable growing conditions, *Daphnia* parthenogenetically produce offspring that build up a population consisting of only females, resulting in exponential growth of clonal populations. On the other hand, under unfavorable conditions, males are produced by parthenogenesis (ESD) and the reproductive mode changed to sexual reproduction; this means that both females and males share the same genome information. Sexually produced eggs, commonly called resting or ephippial eggs, are then formed, which can tolerate extreme conditions (e.g., drying and freezing). These resting eggs can hatch out and develop as females when favorable conditions are restored. In this way, daphnids take advantage of cyclical parthenogenesis depending on changing environmental conditions in their habitat; parthenogenesis allows rapid propagation during favorable growing seasons, whereas sexual reproduction contributes to an increase in genetic variation and survival rate [114].

In terms of sex determination, several studies have demonstrated that various environmental cues such as photoperiod, temperature, nutrition, and crowding, trigger the production of male offspring in *Daphnia* [36,37,112,113]. Despite great efforts in studies on male induction, reproducible experimental conditions for the production of male offspring have not been established yet. However, JHs and their agonists such as methoprene and fenoxycarb have been demonstrated to induce a dose-dependent increase in male offspring in the water flea *Daphnia magna* [115,116,117,118,119,120,121] and other cladoceran species such as *Ceriodaphnia*, *Moina*, *Bosmina*, *Oxyurella*, *Leberis*, *Leydigia*, and *Disparalona* [117,122,123,124]. JHs and their agonists activities can be estimated in vitro by luciferase assays using *Daphnia* JH receptor complex (methoprene-tolerant and steroid receptor coactivator) [125,126,127], and *in silico* by molecular docking simulations between the protein structure of the *Daphnia* methoprene-tolerant and chemicals (e.g., JHs and their agonists) [128]. So far, no one has succeeded in quantifying innate MF levels in extracts from daphnid species. In the near future, quantification of endogenous MF levels during the sex determination period will be indispensable for understanding its physiological role as a male sex determinant. Although JH-induced male production has enabled further studies regarding understanding the molecular mechanisms underlying masculinization processes in daphnids [33,129], the factors responsible for male sexual development are still not well-understood. Recently, our group has identified the *doublesex1* (*dsx1*) gene, which exhibits male-specific expression patterns from early embryonic to adult stages; knockdown of *dsx1* in male embryos and ectopic expression of *dsx1* in female embryos resulted in sex reversed phenotypes, in *D. manga* [130] and in other cladocerans [124] (Figure 2). Recently, components of gene cascade connecting JH signaling to *dsx1* have been identified as bZIP transcription factor, Vrille [131] and the doublesex1 alpha promoter-associated long noncoding RNA (DAPALR) [132].

Our group has recently found a useful *D. pulex* strain (WTN6 strain) that can produce male and female offspring in response to day-length differences: a mother produces female progeny reared under the long-day condition (14 h light, 10 h dark), whereas male progeny emerge under the short-day condition (10 h light, 14 h dark) [33]. This is a suitable experimental tool that enables the evaluation of factors involved with the MF signaling pathway governing ESD in daphnids. Taking advantage of the WTN6 strain, we have successfully identified the male-sex determining factors by transcriptome analysis: ionotropic glutamate receptors, especially N-methyl-D-aspartic acid (NMDA) receptor subtypes, and protein kinase C (PKC) act as upstream regulator of MF signaling and are involved in signaling pathways inducing male offspring [133,134]. Although it has been reported that PKC can recruit NMDA receptors to the cell surface in *Xenopus* oocytes and then increase their channel-opening rates [135], the causal relation between NMDA and PKC pathways for MF signaling in daphnids remains unclear. Likewise, metabolome analysis found that pantothenate (generally known as a vitamin B5) is highly accumulated in individual mothers at the onset of the sex-determining period, when reared under male-producing conditions [136]. Pantothenate is ubiquitously present in living organisms and is known as a precursor of co-enzyme A (CoA). Interestingly, treatment of mother individuals with pantothenate demonstrated that the male induction ratio was significantly increased, suggesting that it may act as a male-sex determinant. So far, however, the role of pantothenate in the activation of MF signaling is largely unknown. One possible hypothesis is that pantothenate can be supplied as a primary source for the MF synthesis pathway, because MF is a member of the sesquiterpenoids that are initially synthesized from acetyl-CoA through the mevalonate pathway. More detailed analyses will be necessary for elucidation of the pantothenate involvement in MF biosynthesis in daphnids. 

To support those findings about MF signaling driving male sex determination in the WTN6 strain more robustly, our group recently found two *D. magna* strains (LRV13.2 and LRV13.5-1 strains) in which the proportion of the female or male offspring can be altered depending on photoperiod: The LRV13.2 strain produces female or male offspring when reared under long-day or short-day conditions, respectively (in a similar manner to the *D. pulex* WTN6 strain), whereas the LRV13.5-1 strain conversely produces female or male offspring reared under short-day or long-day conditions, respectively [137]. Moreover, we clearly confirmed that signaling pathways underlying male sex determination processes are regulated by MF signaling via ionotropic glutamate receptors and PKC pathways in the both the LRV13.2 and LRV13.5-1 strains as well as the WTN6 strain, whereas pantothenate did not show male inducibility, suggesting that male sex determining processes may be diverged between *D. magna* and *D. pulex* [138] (Figure 4).

## 6. Vertebrate-Type Steroid Hormones

In crustaceans and other arthropod species, ecdysteroids are the only known steroid hormone family known to plays a pivotal role in molting and other developmental processes [139]. In Malacostracans, ecdysteroids are synthesized and secreted from the Y-organ which is regulated by sinus gland-derived neuropeptides, such as a molt-inhibiting hormone (MIH) [140,141,142]. However, it has been demonstrated that vertebrate-type sex steroids are involved not only in reproduction [88,143,144], but also in partial disruption of sex differentiation in decapods. In fact, enzyme immunoassays have successfully detected the vertebrate-type steroids, including 17β-estradiol (E2), estriol, progesterone, testosterone, and 11-ketotestosterone, in the hemolymph of kuruma prawn *M. japonicus* [145]. Treatment of female individuals with testosterone resulted in the masculinization of the ovary in the ghost crab *Ocypoda platytarsis* [146]. An apparent bias towards female occurred the freshwater amphipod *Gammarus pulex* [147] and to the pacific white shrimp *L. vannamei* [148] after treatment with E2. Furthermore, transcriptome analysis revealed that E2 may promote female differentiation in the mud crab *S. paramamosain* [149].

As in the Branchiopoda cladoceran water flea, *D. magna,* it is known that the ecdysteroids are the only steroid family in the Malacostracans as well, and the gut has been identified as a candidate organ for ecdysteroidgenesis [150,151]. Several studies have demonstrated that vertebrate-type steroids (e.g., estrogens, testosterones, and progesterone) and their agonists (e.g., diethylstilbestrol, nonylphenol, and bisphenol A as estrogen agonists, and R-1881 as an androgen agonist) can affect the growth rate, fecundity and entire sex ratio of a population [152,153,154,155,156,157]. However, there are some inconsistencies in these results caused by different experimental procedures. It will be necessary to re-survey the *in vivo* effects of these vertebrate-type steroids on *Daphnia* using widely-accepted validated procedures such as the OECD Test Guideline 211 ANNEX7, “*Daphnia magna* Reproduction Test” [158]. In addition, we have successfully constructed a two-hybrid system using the *D. magna* ecdysone receptor and its heterodimeric partner ultraspiracle complex (EcR/USP) [159], allowing the observation of dose-dependent activation of the EcR/USP when transfectants are exposed to ecdysteroids and other chemicals known to have ecdysteroid-like activities in vitro. Although it will be necessary to check the cross-reactivity of vertebrate-type steroids to *D. magna* EcR/USP, this system can be a useful tool for rapid screening, instead of *in vivo* assays. Moreover, recent progress in big data-driven computational (*in silico*) analysis has enabled the prediction of the interaction of *D. magna* EcR/USP with chemicals [139,160]. This structure-based *in silico* approach is very compatible with *de novo* transcriptomics to build comprehensive gene models even in non-model species, and can be easily applied in various organisms as an efficient and cost-effective tool for screening large inventories of chemicals for their potential to cause endocrine disruption.

## 7. Conclusions and Future Directions

This review serves as an outline reference for endocrine-driven sex determination and/or sexual differentiation systems in Malacostraca and Branchiopoda crustaceans. Although the transcriptional regulatory mechanisms between *IAG* and *Dmrt* genes have been investigated by gene knockdown approaches [161,162], the eyestalk (XO-SG complex)-derived neuropeptides that regulate IAG expression and those molecular networks are still largely unknown. Moreover, even among Malacostraca species, previous findings have so far demonstrated that endocrine systems vary in certain respects such as heterodimeric disulfide bond patterns of IAG. Recent advances in OMICS technologies and genetic manipulation techniques have paved the way for a new generation of research organisms, including crustaceans. Indeed, as fast-growing model crustaceans, the Branchiopoda *Daphnia* (*D. pulex* and *D. magna*) and the Malacostraca amphipod *Parhyale hawaiensis* are useful because these species are easy to rear and offer large broods of embryos amenable to dissection and live imaging, and complete embryonic developmental staging [163,164]. In addition, genome sequences are available [30,34,35]. Microinjection-based genetic manipulation using genome editing combined with draft genome and transcriptome archives have enabled further studies of evolution and development in arthropods [163,165,166,167]. However, the decapod species have no established and widely-accepted model species, despite their importance for fisheries and aquaculture. Although the cherry shrimp *N. denticulate* and the parthenogenetic marbled crayfish *P. fallax f. virginalis* are available for developmental and physiological studies with genome sequences and offer useful experimental advantages [25,26,168,169], genomic manipulation methods have not been established so far. As more researchers continue to adopt decapods (and other crustaceans) into their laboratories and study their endocrinology and continue to develop genomic manipulation methods, it will be exciting to see the new research horizons of not only sexual development, but also unexpected phenomena with this unique emerging research system. In summary, for future sex determination/differentiation studies in crustaceans, establishment of useful model crustacean (especially decapod) species and reverse genetics methods will be essential.

## Figures and Tables

**Figure 1 genes-12-00305-f001:**
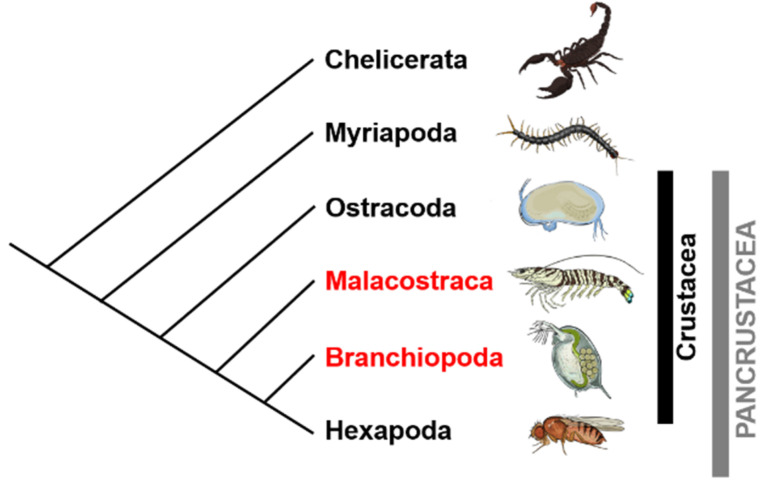
Phylogeny of extant arthropods. The branching pattern is based on [1] with some modifications. Crustaceans are one of the four major groups of arthropods and consist of Ostracoda, Malacostraca, and Branchiopoda.

**Figure 2 genes-12-00305-f002:**
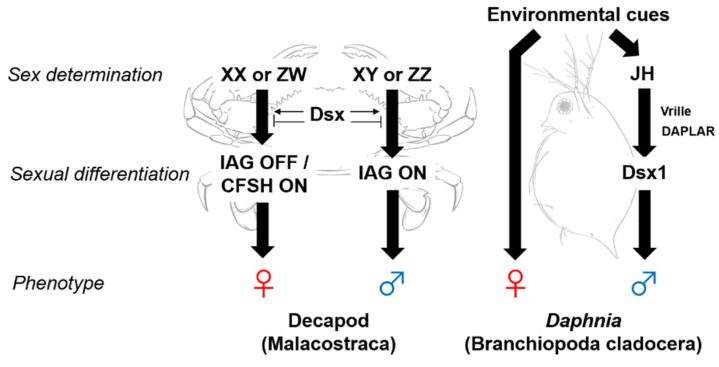
Schemes of sex determination and sexual development of decapods (Malacostraca) and the water flea *Daphnia* (Branchiopoda, cladoceran). IAG: insulin-like androgenic gland hormone, CFSH: crustacean female sex hormone, JH: juvenile hormone, DAPALR: doublesex1 alpha promoter-associated long noncoding RNA, Dsx: doublesex.

**Figure 3 genes-12-00305-f003:**
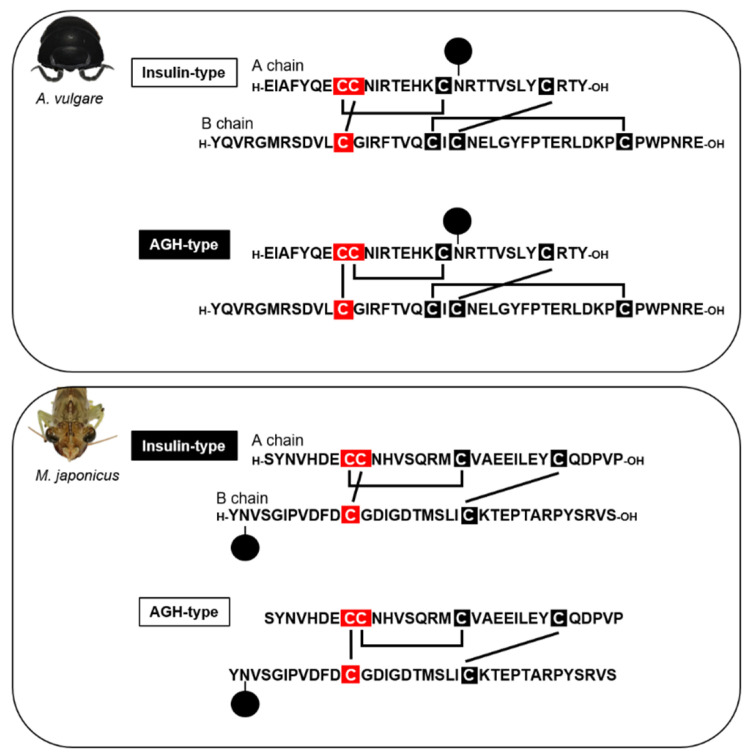
Primary structures of IAG peptides in the woodlouse *A. vulgare* (upper) and the kuruma prawn *M. japonicus* (lower). Each upper and lower shows the insulin-type and AGH-type, respectively. Solid lines show the disulfide bond pairs, and cystein residues in the red box indicate the different patterns between insulin-type and AGH-type. Black-highlighted types (AGH-type in *A. vulgare* and insulin-type in *M. japonicus*) are the estimated bioactive forms, respectively. ●: sugar chain.

**Figure 4 genes-12-00305-f004:**
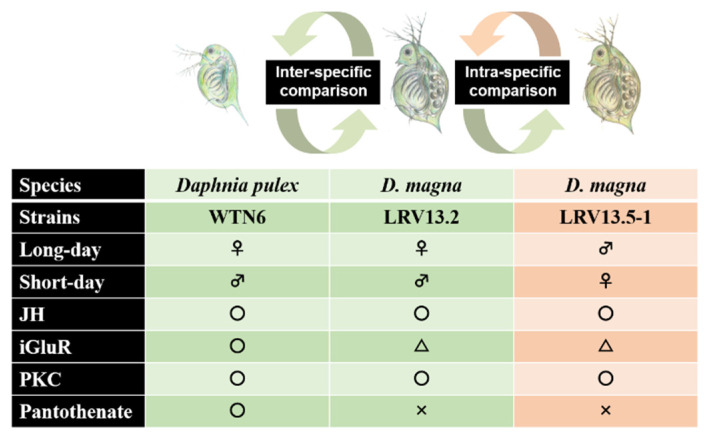
Comparative schematic diagram of putative molecular signaling cascades regulating male offspring production in *D. pulex* WTN6 strain (**left**), in *D. magna* LRV13.2 strain (**center**), and in *D. manga* LRV13.5-1 strain (**right**). Long-day: 14 h-light and 10 h-dark, short-day: 10 h-light and 14 h-dark, JH: juvenile hormone, iGluR: ionotropic glutamate receptor, PKC: protein kinase C. Each symbol (circle, triangle, and cross) mean the “male-inducible”, “non male-inducible”, “treatment of antagonists can suppress male induction, but that of agonists cannot”.

**Table 1 genes-12-00305-t001:** Modes of sex determination and available genome in Malacostracan and Branchiopod crustaceans.

Species	Taxonomy	Sex Determination Manner	Draft or Complete Genome
Pacific white shrimp *Litopenaeus vannamei*	Malacostraca Decapoda	GSD with ZZ/ZW [21]	[22]
Giant freshwater prawn *Macrobrachium rosenbergii*	Malacostraca Decapoda	GSD with ZZ/ZW [23]	[24]
Cherry shrimp *Neocaridina denticulate*	Malacostraca Decapoda	Not available	[25]
Marbled crayfish *Procambarus fallax f. virginalis*	Malacostraca Decapoda	GSD (no male has reported)	[26]
Mud crab *Scylla paramamosain*	Malacostraca Decapoda	GSD with ZZ/ZW [27]	[28]
Wood louse *Armadillidium vulgare*	Malacostraca Isopoda	GSD with ZZ/ZW [9,10]	[29]
*Parhyale hawaiensis*	Malacostraca Amphipoda	Not available	[30]
*Gammarus duebeni*	Malacostraca Amphipoda	ESD [31]	Not available
Water flea *Daphnia pulex*	Branchiopoda Cladocera	ESD [32,33]	[34,35]
Water flea *Daphnia magna*	Branchiopoda Cladocera	ESD [36,37]	[38]
Clam shrimp *Eulimnadia texana*	Branchiopoda Spinicaudata	GSD with androdioecious (male and hermaphrodite) [39]	[40]

GSD and ESD indicate genotypic and environmental sex determination, respectively.

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
