# Peer review of "Sex Determination and Differentiation in Decapod and Cladoceran Crustaceans: An Overview of Endocrine Regulation"

_genes, 2021, doi:10.3390/genes12020305_

Round 1
Reviewer 1 Report
In this manuscript, the author aimed to review the Crustacean sex determination and sex differentiation, with an overview on the endocrine regulation mechanisms involved in such species. This is an interesting and valuable study which encompass several knowledge on the sex determination systems and the associated sex differentiation pathways occurring in Decapods and Branchiopods (Crustacean). The manuscript is generally clear, well organized and well written. I do not have any major concern on the information reviewed and provided, however I do have some minor concerns on some terms and bibliographical references used.
First of all, I have a question concerning the title itself of the article. This one invites the reader to read a review concerning all crustaceans, while the article focuses essentially (see only) on the knowledge present in branchiopods and decapods. What about Amphipoda, Isopoda, Mysida, etc.? I fully understand and hear that it is difficult to summarize so much knowledge in an article (especially since much information is quite well known and described in other groups), so I recommend to change the title of the article, as it makes very few references to other phyla than Branchiopoda/Decapoda. To me, the actual title is not truly representative of the article content.
Line 92/93: The authors used the ref 9 to exemplify the ZZ/ZW system occurring in isopods. However this reference is not appropriate, since it concern only one species of isopod (a group known to have both ZW and XY systems, depending on the species), and is not focused on the sex determination mechanism of this species. Please cite Juchault and Rigaud 1995 and/or Becking et al 2017 instead.
Juchault, P.; Rigaud, T. Evidence for Female Heterogamety in Two Terrestrial Crustaceans and the Problem of Sex Chromosome Evolution in Isopods. Heredity 1995, 75, 466–471.
Becking, T.; Giraud, I.; Raimond, M.; Moumen, B.; Chandler, C.; Cordaux, R.; Gilbert, C. Diversity and Evolution of Sex Determination Systems in Terrestrial Isopods. Scientific Reports 2017, 7, 1084, doi:10.1038/s41598-017-01195-4.
Line 117/121: The authors explained that several genomes of crustacean are now available in public databases, citing 5 of them. I think it could be better to include all the genomes available (in NCBI for example), since a bit less than 50 are now published, especially since few of them were sequenced associated with works on sex determination. May be using a table, a supplementary table, etc. And if not, why citing those 5 genomes instead of others?
Line 204/207 (and in general): I am quite surprised to not see any mention of the associated molecular receptors to AGH-type molecules, such as circulating or transmembrane receptors (insulin-like growth factor-binding proteins, etc.). May be it should be valuable to add some information on it in the manuscript, since these molecules are highly correlated with the activity and the effectiveness of the AGH-type molecules.
Author Response
First of all, I have a question concerning the title itself of the article. This one invites the reader to read a review concerning all crustaceans, while the article focuses essentially (see only) on the knowledge present in branchiopods and decapods. What about Amphipoda, Isopoda, Mysida, etc.? I fully understand and hear that it is difficult to summarize so much knowledge in an article (especially since much information is quite well known and described in other groups), so I recommend to change the title of the article, as it makes very few references to other phyla than Branchiopoda/Decapoda. To me, the actual title is not truly representative of the article content.
Reply: Thank you for your suggestion. Following your kind suggestion, we revised the title as “Sex determination and differentiation in decapod and cladoceran crustaceans: an overview of endocrine regulation”.
Line 92/93: The authors used the ref 9 to exemplify the ZZ/ZW system occurring in isopods. However this reference is not appropriate, since it concern only one species of isopod (a group known to have both ZW and XY systems, depending on the species), and is not focused on the sex determination mechanism of this species. Please cite Juchault and Rigaud 1995 and/or Becking et al 2017 instead.
- Juchault, P.; Rigaud, T. Evidence for Female Heterogamety in Two Terrestrial Crustaceans and the Problem of Sex Chromosome Evolution in Isopods. Heredity 1995, 75, 466–471.
- Becking, T.; Giraud, I.; Raimond, M.; Moumen, B.; Chandler, C.; Cordaux, R.; Gilbert, C. Diversity and Evolution of Sex Determination Systems in Terrestrial Isopods. Scientific Reports 2017, 7, 1084, doi:10.1038/s41598-017-01195-4.
Reply: Thank you for your kind suggestion. We revised citations as suggested.
Line 117/121: The authors explained that several genomes of crustacean are now available in public databases, citing 5 of them. I think it could be better to include all the genomes available (in NCBI for example), since a bit less than 50 are now published, especially since few of them were sequenced associated with works on sex determination. May be using a table, a supplementary table, etc. And if not, why citing those 5 genomes instead of others?
Reply: Thank you for your valuable comments. We added a table entitled “Modes of sex determination and available genome in Malacostracan and Branchiopod crustaceans.” as Table 1. It includes sex determination manner and available genome information of representative crustaceans including dacapods, isopods, amphipod, and branchiopods. As you pointed out, to date a much of genome information of crustaceans is available, however, we realize it is difficult to connect between existence of complete genome sequence and identification of sex-determining genes, even in the sex chromosome identified species. Because identification of sex-determining genes also requires other technical factors such as availability of the organisms (e.g., ease of breeding) and of reverse genetics approaches. Therefore, here, we just mentioned current knowledge of available “representative” crustacean genome. Additionally, we described several examples of sex-specific transcriptome studies for more useful information of sex determining and/or differentiation research. In lines 123-128, we revised as “Especially genome information of following species: the marbled crayfish Procambarus fallax f. virginalis [26], the Pacific white shrimp Litopenaeus vannamei [22], the giant fresh-water prawn Macrobrachium rosenbergii [24], and the terrestrial isopod Armadillidium vul-gare [29], will accelerate sex determination and differentiation studies, since they have been used in the studies of endocrinology and sex differentiation as experimental animals (details in following sections).”.
Line 204/207 (and in general): I am quite surprised to not see any mention of the associated molecular receptors to AGH-type molecules, such as circulating or transmembrane receptors (insulin-like growth factor-binding proteins, etc.). May be it should be valuable to add some information on it in the manuscript, since these molecules are highly correlated with the activity and the effectiveness of the AGH-type molecules.
Reply: As suggested, in lines 189-190, we added a sentence about IAG binding protein and receptor with appropriate references.
Reviewer 2 Report
In this review, the authors summarized current knowledge on mechanism of sex determination and differentiation in crustaceans, including GSD and ESD species. It provides substantial insights for the future research. This paper is well organized and written.
Author Response
Thank you so much for your reviewing.